# Mixotrophy for carbon-conserving waste upcycling

**Michael Weldon** **, Christian Euler** *

Department of Chemical Engineering, University of Waterloo, Waterloo, Ontario, Canada

* ceuler@uwaterloo.ca

**Data availability statement:** All data and code used for analysis and visualization are available at https://github.com/theL-A-B.

**Funding:** This work was supported by the National Sciences and Engineering Research

## Abstract

Modern chemical manufacturing, on which human quality of life depends, is unsustainable; alternative production routes must be developed. Electrochemical and biological processes offer promise for upgrading waste streams, including recalcitrant carbon dioxide and plastic-derived wastes. However, the inherent heterogeneity and high energy requirements of upcycling the chemical endpoints of the "take-make-waste" economy remain challenging. *Cupriavidus necator* is an emergent catalyst for complex feedstock valorization because of its extreme metabolic flexibility, which allows it to utilize a wide array of substrates, and its ability to use carbon dioxide via the Calvin-Benson-Bassham cycle. *C. necator* natively oxidizes hydrogen to power carbon utilization, but its flexibility offers an as-yet unexplored opportunity to couple waste stream oxidation with carbon dioxide utilization instead, potentially enabling carbon conservative waste upcycling. Here, we uncover the constraints on carbon conservative chemical transformation using *C. necator* as a model. We systematically examine the carbon yield and thermodynamic feasibility of mixotrophic scenarios combining waste-derived carbon sources with hydrogen oxidation to power carbon reassimilation. Then, we evaluate carbon-carbon mixotrophic scenarios, with one carbon source providing electrons in place of hydrogen oxidation. We show that both hydrogen and ethylene glycol are feasible electron sources to drive carbon-neutral or carbon-negative mixotrophic upgrading of waste streams such as acetate or butyrate. In contrast, we find that carbon conservation is likely infeasible for most other waste-derived carbon sources. This work provides a roadmap to establishing novel *C. necator* strains capable of carbon efficient waste upcycling.

## Author summary

Decarbonizing chemical manufacturing is critical to achieving a sustainable future, and bioprocesses will play an important role in this pursuit. *Cupriavidus necator* is a promising microbial platform for chemical production from carbon dioxide, which it natively uses to make polyesters. However, *C. necator* uses hydrogen to drive carbon dioxide utilization, which poses some practical challenges to its use as an industrial catalyst.

Council of Canada (ALLRP 597254-24 to CE). The funders had no role in study design, data collection and analysis, decision to publish, or preparation of the manuscript.

**Competing interests:** The authors have declared that no competing interests exist.

Green hydrogen production is energy intensive, hydrogen carries significant safety risks, and gas transfer limitations in bioreactors may be difficult to overcome. We consider coupling other sustainable electron sources to carbon utilization in *C. necator* to understand whether waste-derived feedstocks could support carbon neutral or even carbon negative polymer production. We evaluate a number of feedstocks that can be derived from plastic depolymerization, electrochemical carbon dioxide reduction, and other fermentation processes for their ability to support carbon conservation in mixotrophic scenarios. Our stoichiometric and thermodynamic analysis shows that alternative electron sources could improve carbon utilization and are subject to the same metabolic constraints as carbon dioxide/hydrogen mixotrophic fermentation. This work highlights the feasibility of using *C. necator* to upgrade multiple waste streams, including carbon dioxide, simultaneously, and identifies the potential bottlenecks to doing so.

## Introduction

Today, chemical production is heavily reliant on petroleum: thousands of chemical products begin as petroleum, and much of the energy used to produce them is also petroleum-derived [1,2]. Continued long-term reliance on this carbon source is impossible because its accessible reserves are predicted to run out before the end of the century [3–5]. Carbon dioxide is an attractive replacement because it can be sustainably sourced as part of the global carbon cycle and its ubiquity as a product of human industrial activities could make its use cost-effective. To date, the primary strategies for carbon dioxide upgrading have been electrochemical and biological.

In electrochemical reduction of carbon dioxide ($eCO_2R$), electrical energy is used to convert carbon dioxide and a hydrogen/electron donor - water or hydrogen - into value-added chemicals. The sustainability of such electroreduction processes depends on the sustainability of the energy generation methods used to power them [6]. Power grids are unlikely to be able to quickly accommodate the energy demands of large-scale chemical production via $eCO_2R$ without support from the continued burning of fossil fuels for electricity generation [7]. Broad adoption of carbon dioxide upgrading will therefore require massive expansion of renewable energy generation alongside efficiency improvements to existing renewable energy technologies [7,8].

In addition to these significant energy requirements, $eCO_2R$ also requires a large volume supply of hydrogen that must be sourced sustainably. Green hydrogen is often considered as a viable way to meet this demand, but the required technology for its production has not been demonstrated at an industrial scale to date [9,10]. It will thus be a significant challenge to meet the 250 Mt green hydrogen demand that is predicted by 2050. At this scale, the generation of this input stream alone would use up approximately 90% of the energy budget of the chemical sector. Moreover, increasingly limited water supplies in a changing climate will likely set an upper bound on green hydrogen production capacity [10]. The constraints on sustainable hydrogen production therefore represent a significant barrier to the expansion of $eCO_2R$-based chemical production [8].

Even if these material and energetic barriers are overcome, the scope of products accessible via $eCO_2R$ will have to be expanded for it to be a meaningful approach to meet the current chemical product needs of society. The primary outputs of $eCO_2R$ processes to date have been C1 and C2, and to a lesser extent, C3 chemicals. Carbon monoxide, formic acid, ethylene, methanol, ethanol, methane, and *n*-propanol [11] are the most often reported products. Other C3 chemicals and some C4 chemicals have been observed in $eCO_2R$ processes but at

limited production rates and scales [12–14]. Part of the limitation to producing longer carbon chain molecules is that the C1/C2 intermediates are released from the catalyst before they can be extended [14]. As such, producing polymers and liquid fuels from carbon dioxide remains challenging, though work toward achieving sustainable conversion processes is accelerating [15,16].

In contrast to electrochemical production methods, biological routes leverage the native ability of certain microbes to fix carbon dioxide as a way to produce value-added compounds. For example, acetone and isopropanol (C3 compounds) production by engineered *Clostridium autoethanogenum* grown on waste gas feedstocks (carbon dioxide/carbon monoxide/hydrogen) has been demonstrated at industrial scale with productivities reaching approximately 3 g/L/h and selectivities of up to 90% [17]. Anaerobic acetogens, such as *C. autoethanogenum*, are attractive for carbon dioxide utilization because they make use of the energy efficient Wood Ljungdahl pathway [18,19] and do not require intensive process development for light delivery, as is the case with photosynthetic organisms [20,21]. However, the strict anaerobic requirements of acetogens might limit the scope of products that is accessible via processes utilizing them in two ways. First, it stoichiometrically constrains oxygen availability, because oxygen can only come from the carbon dioxide/monoxide feedstock in such processes. Second, it constrains the amount of energy available for the oxidation half of redox reactions, which puts an upper bound on the rate of carbon dioxide reduction.

The emerging industrial platform organism *Cupriavidus necator* can instead fix carbon dioxide in the presence of oxygen using a Calvin-Benson-Bassham (CBB) cycle powered by unique, oxygen-tolerant hydrogenases [22,23]. *C. necator* has been used to produce a wide range of compounds, such as alcohols, fatty acids, terpenes, alkanes/alkenes, and organic acids [24]. Its ability to grow in the presence of oxygen, coupled with its inherent metabolic flexibility, makes it attractive as a host to oxidize alternative, non-sugar carbon sources, such as those derived from waste [25]. Furthermore, its ability to fix carbon in the presence of oxygen presents a significant opportunity to engineer the energetic coupling between oxidation pathways and carbon dioxide reduction. For example, the oxidation of $eCO_2R$-derived C1 substrates such as formate or methanol could provide energy in addition to that available from hydrogen to drive carbon dioxide fixation. However, there are a number of other waste products that could potentially be used as sources of electrons to power carbon dioxide fixation in *C. necator*. Plastic monomers are of particular interest, because of the accelerating plastic crisis [26] and because *C. necator* natively produces polyhydroxybutyrate (PHB), a biopolymer that could be used as an alternative to petroleum-based plastics [27].

The idea of using mixotrophic approaches to increase carbon yield in microbial metabolism is not new, but has not been explored extensively or systematically. Previous in silico work evaluated mixotrophic strategies combining methanol oxidation with the transformation of various sugars via non-oxidative glycolysis (NOG). This demonstrated that methanol oxidation can increase the thermodynamic driving force for the production of desired chemicals, while conserving carbon [28]. Anaerobic carbon-conserving mixotrophy has been experimentally established in *C. autoethanogenum* engineered to utilize sugars and carbon dioxide [29] and *Clostridium carboxidivorans* engineered to use sugars and carbon monoxide [30]. In both cases, yields were increased because reducing equivalents from sugar oxidation could be used to drive C1 reduction. Similarly, *C. necator* has been used to upgrade mixed streams of volatile fatty acids with yield increases demonstrated when carbon fixation pathways were activated by culture conditions, but with limited demonstration of mixotrophic metabolism [31].

More broadly, recent work has introduced the idea of rewiring metabolism to couple fermentative and respiratory pathways, with successful experimental implementation in an

obligate anaerobic *Escherichia coli* strain and demonstration of carbon conservation [32]. In another instance, sugar-formate mixotrophy was achieved in engineered *Yarrowia lipolytica*, with metabolically-derived carbon dioxide being electroreduced and fed back into the fermentation as formate to conserve carbon [33]. Promisingly, it was very recently demonstrated that *C. necator* engineered to constitutively express CBB pathway genes can readily take up carbon dioxide to address redox imbalances when it is fed palmitate, resulting in significant carbon efficiency increases [34]. Together, these examples demonstrate the validity and potential of mixotrophic use of carbon dioxide-derived feedstocks to reduce the carbon intensity of fermentation.

Motivated by this previous work, and by the unique potential offered by the native ability of *C. necator* to fix carbon aerobically, we establish an in silico workflow here to systematically evaluate the stoichiometric and thermodynamic constraints on mixotrophic scenarios making use of waste-derived, chemically reduced inputs. We initially consider coconsumption of carbon feedstocks which could be derived from waste streams with hydrogen as an electron donor. Then, we consider carbon-carbon mixotrophies in various configurations to uncover feasible, carbon-conserving routes to polymer and acid production that do not rely on hydrogen-derived electrons. We show that such mixotrophies are thermodynamically favourable and identify those which improve carbon efficiency, providing a route to the design of novel microbial systems capable of efficiently upcycling waste-derived feedstocks.

## Results

Transformation of carbon dioxide to value-added products is a net chemical reduction process; therefore, it must be coupled to a complementary oxidation process. Based on the native ability of *C. necator* to couple hydrogen oxidation with carbon dioxide reduction, we aimed to evaluate the feasibility of coupling the oxidation of reduced, waste-derived carbon sources with carbon dioxide reduction instead (Fig 1A), supposing that such coupling could decrease or eliminate the need for electrons from hydrogen. We considered reduced carbon sources available via three streams: plastic degradation and/or depolymerization; existing eCO$_2$R processes; and byproducts of bioprocesses. We further limited our search to carbon sources with known assimilation pathways, whether native to *C. necator* or not.

This yielded a list of ten possible substrates: formate, succinate, acetate, adipate, glycerol, butyrate, ethylene glycol (EG), 1,4-butanediol (14BDO), ethanol (EtOH), and methanol (MeOH). Possible sustainable sources of these substrates, along with their assimilation pathways, are presented in Table 1. Ethylene glycol is unique among the substrates we considered in that it could be derived from all three streams considered. We also included carbon dioxide and fructose as benchmark carbon sources for comparison, since these are native substrates for *C. necator*. We added reported assimilation pathways for each of these carbon sources individually to a core model of *C. necator* metabolism [35] modified to include hydrogen oxidation (Fig 1B) to yield eleven models capable of consuming non-carbon dioxide carbon sources plus one which can only use carbon dioxide/hydrogen. For details about the core metabolic model, including modification and validation, see Materials and Methods.

### Highly reduced substrates require less hydrogen for complete carbon conversion

Our focus was on the feasibility of various input scenarios, so we used PHB as a common product for comparison across the substrates evaluated here. PHB production envelopes for all models with both complete carbon conversion (i.e. without loss to carbon dioxide) supported by hydrogen-derived electrons and without forced complete conversion, generated

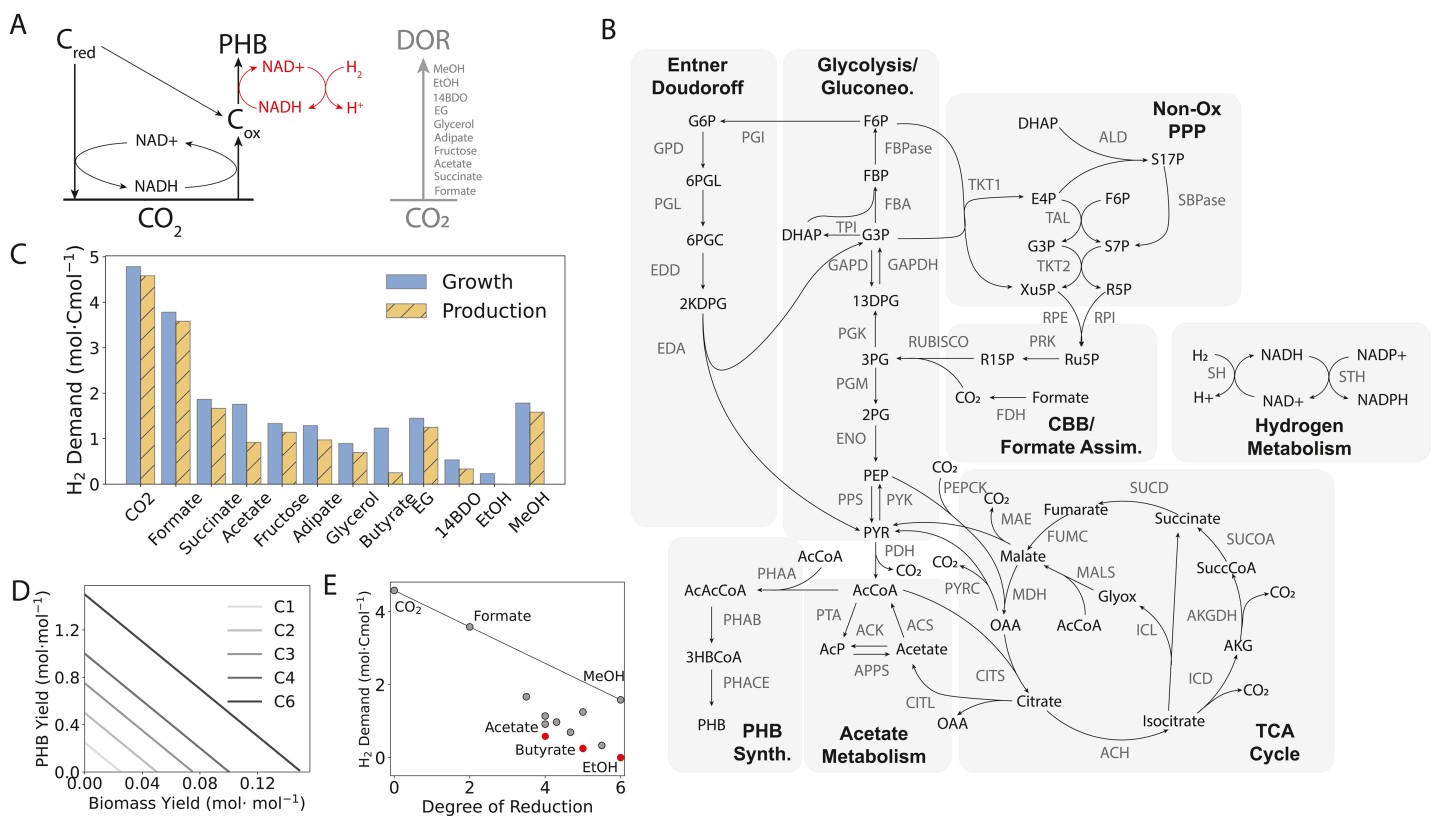

**Fig 1. (A) Schematic of redox coupling using alternative, reduced substrates.** Energy derived from the oxidation of reduced carbon sources ($C_{red}$) could be coupled to carbon dioxide reassimilation in *C. necator*, with additional energy for transformation to the product polyhydroxybutyrate (PHB) provided by hydrogen oxidation (shown in red). Partial oxidation of high DOR substrates may yield intermediates ($C_{ox}$) for PHB production. Carbon sources with a higher DOR can produce more reducing equivalents for carbon reassimilation, reducing the hydrogen cost to achieve carbon conservation. (B) Map of the core metabolic model used in this work. This model was adapted from [35] with the addition of a soluble hydrogenase (SH) and a membrane-bound hydrogenase that can harvest reducing equivalents from hydrogen. The model has 79 reactions and 68 species. CBB: Calvin-Benson-Bassham cycle. See Supplemental Material for complete metabolite and reaction coding. (C) Hydrogen cost per assimilated carbon mole (Cmol) for growth-only and PHB production-only modes for all carbon sources investigated. All carbon dioxide generated via substrate oxidation was forced to be reassimilated with required reducing equivalents provided by hydrogen oxidation. (D) Production envelopes for carbon conservative fermentation. For all carbon sources, forced reassimilation of carbon dioxide results in stoichiometric conversion (see Fig S1B), so production envelopes depend only on carbon number (C*n*). (E) Hydrogen cost for complete carbon conversion to PHB as a function of degree of reduction (DOR) for all carbon sources. A line is drawn to show the expected hydrogen cost if all carbon were fully oxidized to carbon dioxide then reassimilated. Carbon sources falling below the line are more stoichiometrically efficient than an equivalent C1 substrate. Acetate and ethanol are the most hydrogen-efficient substrates for complete carbon conversion to PHB. Relative hydrogen savings for all substrates are presented in Fig S2.

using parsimonious FBA (pFBA) are plotted in S1B Fig. To enforce complete carbon conversion, RuBisCO and hydrogen uptake flux were unconstrained, while carbon dioxide efflux was constrained to be zero. In the incomplete conversion case, carbon reassimilation was blocked by setting hydrogen uptake to zero and allowing unconstrained exchange of carbon dioxide. The production envelopes show that, unsurprisingly, complete carbon conversion supported by hydrogen oxidation improves both biomass and PHB yield for all operating points relative to the case with no reassimilation.

Moreover, complete conversion removes stoichiometric constraints for all C2+ carbon sources, enabling stoichiometric conversion to biomass and/or PHB for all carbon sources. As a result, production envelopes for the hydrogen-supported complete conversion case are defined solely by the carbon number of the substrate, as shown in Fig 1D. In contrast, when

**Table 1. Consumption modules used to construct C. necator models capable of consuming alternative substrates.** See supplemental material for reaction codes.

| Carbon Source | DOR | Assimilation Reactions | Ref. | Source(s) |
|---|---|---|---|---|
| Formate | 2 | FDH: formate + NAD+ + $H_2O$ = $CO_2$ + NADH | [35] | $eCO_2R$ |
| Succinate | 3.5 | SUCD[1]: succinate + ubiquinone = fumarate + ubiquinol | [35] | Sugar fermentation Polybutylene succinate depol. |
| Acetate | 4 | ACS[1]: acetate + CoA + ATP = acetyl-CoA + PPi + AMP<br>ACK[1]: acetate + ATP = ADP + acetyl-phosphate<br>PTA: acetyl-CoA + Pi = acetyl-phosphate + CoA | [35] | Acetogenic fermentation |
| Adipate | 4.33 | ACL: adipate + ATP + CoA = adipyl-CoA + ADP + Pi<br>ACDH: adipyl-CoA + ubiquinone = 5-carboxy-2-pentenoyl-CoA + ubiquinol<br>CEDH: 5-carboxy-2-pentenoyl-CoA + $H_2O$ = 3-hydroxyadipyl-CoA<br>HADH: 3-hydroxyadipyl-CoA + NAD+ = 3-ketoadipyl-CoA + NADH<br>OAT: 3-ketoadipyl-CoA + CoA = acetyl-CoA + succinyl-CoA | [54] [55] | Sugar fermentation Polybutylene adipate terephthalate depol. |
| Glycerol | 4.67 | GLYK: glycerol + ATP = glycerol-3-phosphate + ADP<br>GKDH: glycerol-3-phosphate + ubiquinone = dihydroxyacetone phosphate + ubiquinol<br>GLDH[2]: glycerol = 3-hydroxypropanal + $H_2O$<br>HPADH[2]: 3-hydroxypropanal + NADP+ + $H_2O$ = 3-hydroxypropionate + NADPH | [44] [56] | Biodiesel side stream |
| Butyrate | 5 | BCL: butyrate + ATP + CoA = AMP + butyryl-CoA + PPi<br>BCDH: butyryl-CoA + FAD = crotonyl-CoA + $FADH_2$<br>3HBDH: crotonyl-CoA + $H_2O$ = (R)-3-hydroxybutyryl-CoA<br>BOX: butyryl-CoA + FAD + NAD + $H_2O$ + CoA = 2 acetyl-CoA + $FADH_2$ + NADH | [57] [58] | Anaerobic digestion |
| Ethylene glycol | 5 | GAR: ethylene glycol + NAD+ = glycoaldehyde + NADH<br>GAD: glycoaldehyde + NAD+ + $H_2O$ = glycolic acid + NADH<br>GDH: glycolic acid + ubiquinone = glyoxylate + ubiquinol<br>GCL: 2 glyoxylate + $H_2O$ = tartronate semialdehyde + $CO_2$<br>TSR: tartronate semialdehyde + NADH = NAD+ + glycerate<br>G2K: glycerate + ATP = 2-phosphoglycerate + ADP | [42] | Sugar fermentation side stream $eCO_2R$ Polyethylene terephthalate depol. |
| 1,4-Butanediol | 5.5 | ADH: 1,4-butanediol + NAD+ = 4-hydroxybutyraldehyde + NADH<br>BDH: 4-hydroxybutyraldehyde + CoA + NAD+ = 4-hydroxybutyryl-CoA + NADH<br>COAT: 4-hydroxybutyryl-CoA + acetate = acetyl-CoA + 4-hydroxybutyrate<br>HBD: 4-hydroxybutyrate + NAD = succinate semialdehyde + NADH<br>SSADH: succinate semialdehyde + NAD + CoA = NADH + succinyl-CoA | [59] | Polybutylene adipate terephthalate depol. |
| Ethanol | 6 | EDG: ethanol + NAD+ = acetaldehyde + NADH<br>ACEDH: acetaldehyde + NAD+ + CoA = acetyl-CoA + NADH | [60] | $eCO_2R$ |
| Methanol | 6 | METDH: methanol + NAD+ = formaldehyde + NADH<br>FADH: formaldehyde + NAD+ + $H_2O$ = formate + NADH<br>FDH: formate + NAD+ + $H_2O$ = $CO_2$ + NADH | [61] [62] | $eCO_2R$ |

[1] - reaction is already present in the core *C. necator* model

[2] - required for the glycerol-to-3HP scenarios only

some carbon reassimilation is forced with hydrogen uptake blocked - i.e. with all reducing equivalents derived from the carbon source alone - biomass and PHB yield decline with increasing carbon conservation (S1A Fig). This may help to explain why carbon dioxide reassimilation is not observed to a significant degree in *C. necator* cells grown on fructose or succinate without hydrogen [36], even though those cells express the enzymes required for assimilation [37].

Hydrogen-supported complete carbon conversion is driven by a mixture of reducing equivalents derived from the carbon source and hydrogen (Fig 1A), so we expected that carbon sources with a high degree of reduction (DOR) would require less hydrogen for complete conversion. Higher DOR means more electrons per carbon atom to produce the necessary reducing equivalents for carbon dioxide reassimilation. In Fig 1C, the hydrogen cost of complete conversion per mole of carbon converted is presented for both growth-only and production-only extremes. As expected, there is a negative correlation between hydrogen demand and DOR for both growth and production modes on a per-carbon basis, as shown in Fig 1E.

The line in this figure shows the cost of complete carbon conversion in the case of full oxidation to carbon dioxide, followed by reduction to PHB, as is required for C1 carbon sources. All C2+ substrates fall below this line, and their distance from it reflects the relative carbon efficiency of their respective conversion pathways to PHB. Thus, all C2+ feedstocks evaluated here are stoichiometrically more efficient than C1 feedstocks, regardless of DOR, and some are moreso than others. S2 Fig captures this differential efficiency by showing the reduction in hydrogen costs for the hydrogen mixotrophic scenario relative to conversion of the carbon source to carbon dioxide first for each carbon source. This suggests that it may be generally more efficient to valorize waste streams via specific C2+ intermediates, rather than utilizing C1 streams directly.

In the case of acetate, butyrate, and ethanol, carbon dioxide reassimilation is not required to produce PHB with full carbon conversion, since these substrates enter metabolism via acetyl-CoA, which is the starting point of the PHB pathway (Fig 2C, S3 Fig). PHB production does require reducing equivalents that the conversion of acetate and butyrate to acetyl-CoA does not sufficiently provide, so carbon dioxide-free conversion requires electrons from hydrogen. However, the proximity of these substrates to acetyl-CoA renders their hydrogen cost per carbon-mole relatively low among the carbon sources examined (Fig 1E, S2 Fig). Carbon dioxide-free conversion of ethanol requires no hydrogen (Fig 1C), because its oxidation to acetyl-CoA yields more reducing equivalents per carbon atom than are required to assimilate these carbon atoms. Thus, from a purely stoichiometric perspective, acetate, butyrate, and ethanol are ideal sole carbon sources for sustainable, carbon dioxide-free PHB production.

These results suggest that if the native metabolic machinery of *C. necator* could be repurposed to reassimilate carbon generated in waste stream upcycling pathways using an additional source of electrons, then higher biomass and product yields could be achieved relative to net carbon dioxide-producing pathways. However, *C. necator* has been observed to natively reassimilate carbon toward complete carbon conversion only in marginal, nutrient-limited, and transitional situations, even when hydrogen is available [38]. Based on this empirical evidence, it is not clear that carbon reassimilation is feasible, even if it would increase yield.

## Carbon dioxide-free fermentation via hydrogen mixotrophy is broadly feasible

We therefore evaluated the thermodynamic feasibility of hydrogen-supported complete carbon conversion for each carbon source by finding the associated max-min driving forces (MDF), and comparing those against the same for the incomplete conversion case. The MDF is the largest possible thermodynamic bottleneck for a given flux distribution within constraints on metabolite concentrations. Pathways with positive MDFs have bottleneck reactions that are thermodynamically feasible (i.e. with $\Delta_r G < 0$), so they are feasible overall. In contrast, pathways with negative MDFs have bottleneck reactions that are thermodynamically infeasible (i.e. with $\Delta_r G > 0$), so they are infeasible. Moreover, pathways with higher MDFs are more

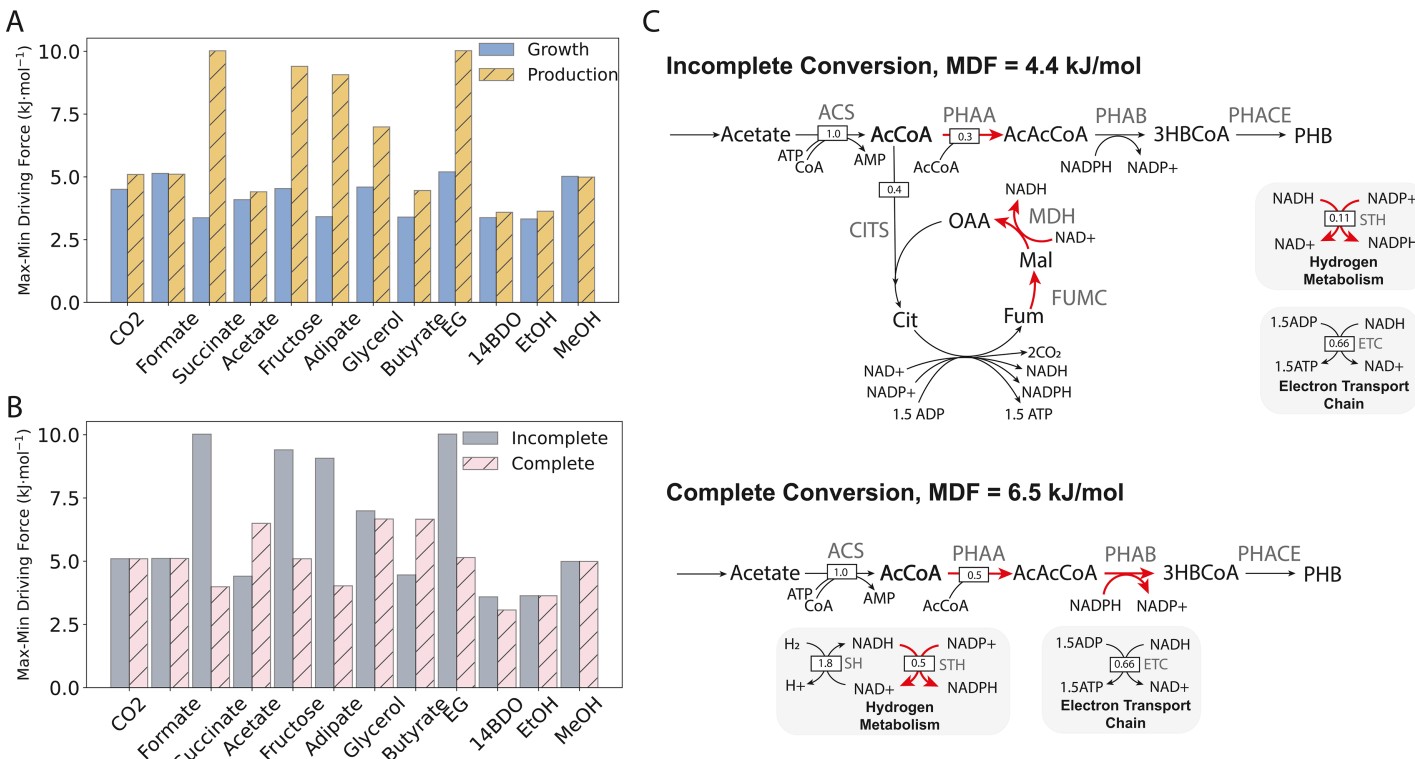

**Fig 2. (A) Maximum-minimum thermodynamic driving force (MDF) for each carbon source for growth-only and production-only extremes without complete carbon conversion (i.e. carbon dioxide excretion allowed).** For all carbon sources except carbon dioxide, hydrogen exchange was blocked so that energy and carbon could be derived only from the given carbon source. Positive MDFs - required for thermodynamic feasibility - are possible for all carbon sources. (B) MDF comparison for PHB production with forced complete carbon conversion ('Complete') and incomplete carbon conversion ('Incomplete'). In the complete conversion case, carbon dioxide excretion was blocked and hydrogen exchange was unconstrained, as in Fig 1C. Complete conversion is thermodynamically feasible for all carbon sources, but incomplete conversion is more thermodynamically favourable for all but acetate and butyrate. (C) Flux distributions for acetate transformation to PHB with incomplete carbon conversion (top panel) and complete carbon conversion (bottom panel) with relative flux indicated. In the unforced scenario, 40% of assimilated acetate must be converted to reducing equivalents via the carbon dioxide-producing TCA cycle. The requirement of fumarase ('FUMC') and malate dehydrogenase ('MDH') activities for energy generation yields an MDF of of 4.4 kJ/mol. All reactions shown in red operate at the MDF. In the complete conversion scenario, hydrogen oxidation provides sufficient reducing power for all of the assimilated acetate to be converted to PHB with no carbon dioxide generation, and therefore no reassimilation required. This reduces the thermodynamic constraint, allowing for operation at an MDF of 6.5 kJ/mol. ACS: acetyl-CoA synthase; PHAA: acetoacetyl-CoA thiolase; PHAB: (R)-3-Hydroxybutanoyl-CoA:NADP+ oxidoreductase; PHACE: PHB synthase; ADK: adenylate kinase; AcCoA: acetyl-CoA; AcAcCoA: acetoacetyl-CoA; 3HBCoA: (R)-3-hydroxybutyryl-CoA.

thermodynamically favourable than those with lower MDFs, with all other things being equal [35,39].

Accurate constraints on metabolite concentrations are critical to determining meaningful MDF values. Constraints on cofactor ratios are especially important in the MDF formulation, since these can significantly change the thermodynamic driving forces for coupled reactions. We modified a standard set of concentration constraints available for *E. coli* [40] to account for the NADH:NAD and NADPH:NADP ratio ranges measured in *C. necator* during mixotrophic fermentation [34]. The modified constraints allow ratios less than one for these cofactors, which is not typical for thermodynamic analysis in other organisms, but captures the ability of *C. necator* to perform net reductive metabolism in mixotrophic scenarios. See Materials and Methods for the MDF formulation and specific constraints used here.

With these cofactor constraints we were able to find optimal flux distributions with positive MDFs for all carbon sources for both growth and production modes without complete

carbon conversion - i.e. with unconstrained carbon dioxide exchange. As shown in Fig 2A, growth MDFs are comparable across carbon sources, indicating that biomass accumulation is bottlenecked by the same set of reactions for each carbon source. Indeed, we found that generally the MDF for growth modes corresponded to operation at the lowest allowed NADH:NAD ratio, since this is required to maximize the driving force for the thermodynamically unfavourable malate dehydrogenase reaction in the TCA cycle ($\Delta G° = 26.5$ kJ/mol) and the marginal transhydrogenase ($\Delta G° = 0$ kJ/mol) and gluconeogenic triose phosphate dehydrogenase ($\Delta G° = -1.2$ kJ/mol) reactions. In contrast, production mode MDFs are variable across carbon sources, indicating variable thermodynamic constraints arising from the specific assimilation pathways of each carbon source. Regardless, production mode MDFs are positive for all yield-optimized pathways, and generally higher than those for the growth mode, indicating broad feasibility of PHB production from all of the substrates evaluated here.

Forcing complete carbon conversion with hydrogen oxidation results in significantly lower MDFs for succinate, fructose, adipate, and ethylene glycol (Fig 2B). This is expected, since maximizing the driving force for carbon fixation requires maximizing the NADH:NAD ratio, whereas maximizing the driving force for substrate assimilation requires minimizing this value to drive coupled dehydrogenase reactions. Despite this, we found feasible, yield-optimized, carbon conserving pathways for these four substrates as well as all the others examined. Unsurprisingly, the C1 substrates all had the same MDF as in the incomplete conversion case, because the same pathways are active in both cases and hydrogen oxidation is highly exergonic, so it will only become a bottleneck with extreme NADH:NAD ratios.

Similarly to the C1 substrates, the MDF for ethanol does not change in the complete conversion case. This is because the two-step conversion to acetyl-CoA does not produce carbon dioxide, but it does produce excess reducing equivalents in the form of NADH. As such, de novo carbon dioxide assimilation is required to balance redox in the conversion of ethanol to PHB (S3A Fig). Even without electrons from hydrogen, the conversion of ethanol to PHB is thermodynamically feasible and carbon dioxide-negative, yielding 0.58 moles of PHB per mole of ethanol assimilated, or 116% of the maximum yield from ethanol alone. However, though ethanol is thermodynamically favourable as a carbon dioxide-negative substrate, its practical application as such is likely limited because it is toxic to microbial cells. Previous work has shown that exposure of *C. necator* to sub-lethal concentrations of ethanol can improve PHB production by inducing redox stress that PHB production helps to balance [41]. As such, supplementation of ethanol may be a viable fermentation strategy, but it is likely not realistic as a primary feedstock for *C. necator*-based processes.

Acetate and butyrate are the only substrates for which complete carbon conversion is more thermodynamically favourable than incomplete carbon conversion. This is because they can both be converted to acetyl-CoA without carbon loss using energy from hydrogen oxidation, provided adenylate kinase (ADK) can recycle ATP to drive initial phosphorylation reactions (Fig 2C, S3A Fig, lower panel). Conversely, in the incomplete carbon conversion case, energy for PHB production is generated via the carbon dioxide-producing TCA cycle (Fig 2D, S3A Fig, upper panel). Thus, fumarase and malate dehydrogenase activities constrain energy generation in the incomplete conversion scenario and this constraint is alleviated by the much more thermodynamically favourable hydrogen oxidation. Unlike ethanol, acetate and butyrate are viable fermentation feedstocks [31], so mixotrophic fermentation with hydrogen should be further considered as a potential carbon dioxide-free path to bioplastic production using them.

## Carbon-carbon mixotrophy can improve carbon yield and thermodynamic feasibility

Coupling an electron source to carbon reduction appears to be generally feasible, so we next aimed to evaluate the feasibility of carbon-carbon mixotrophies making use of the same coupling strategy. Our rationale for doing so lies in the fact that the carbon dioxide generated from the fermentation of $eCO_2R$-derived carbon sources may ultimately be neutral in the global carbon balance. Therefore, mixotrophic scenarios making use of the electrons from these carbon sources could at least be net carbon dioxide-neutral, even if these scenarios require carbon dioxide production at the point of fermentation. We built 45 models capable of using each possible pair of carbon sources by performing a pairwise addition of assimilation modules to each of the sole carbon source models we had generated previously. Then, we evaluated the carbon yield and MDF for PHB production using both carbon sources simultaneously to evaluate the viability of all possible synthetic mixotrophies. Yields and driving forces for all mixotrophic scenarios are presented in Fig 3A, with the sole carbon source scenario along the diagonals.

We reasoned that for a given mixotrophic scenario to be viable, it ought to have the same or higher MDF than either sole consumption scenario; that is, that the mixotrophy should be at least as thermodynamically favourable as sole carbon source consumption. If adding an assimilation pathway reduces the driving force through a bottleneck reaction, or adds a new bottleneck reaction, then using one carbon source at a time would likely be more efficient in terms of protein resource allocation and cofactor ratio balancing to maximize driving force. The mixotrophic carbon yield should also be higher than that of either sole consumption case. For example, the carbon yield of ethylene glycol-formate mixotrophy is 0.56 mol·mol$^{-1}$, which is higher than that of formate alone (0.22 mol·mol$^{-1}$), but not EG alone (0.73 mol·mol$^{-1}$). Thus, transformation of EG alone is more carbon efficient than the mixotrophic scenario, so synthetic ethylene glycol-formate mixotrophy may not be worth pursuing, depending on the source of either or both of the mixotrophic substrates.

Based on the MDF criterion, ten carbon source pairs may be viable. These are highlighted in Fig 3A. Formate, EG, and butyrate all have MDFs in multiple pairings that are at least the same as one of the corresponding sole consumption cases, suggesting that mixotrophy is as thermodynamically favourable as the single carbon source consumption. Formate is thermodynamically viable as a mixotrophic partner with glycerol, butyrate, and ethylene glycol, butyrate is viable with formate, acetate, and fructose, and ethylene glycol is viable with formate, acetate, fructose, glycerol, and butyrate. The viability of multiple ethylene glycol mixotrophies suggests that it could be an ideal carbon-based electron source. Of the mixotrophies we examined, only the fructose-acetate pairing is nearly non-viable, as it has an MDF near zero (0.57 kJ/mol).

When the carbon yields of these mixotrophies are considered, ethylene glycol-acetate, and methanol-acetate emerge as the best mixotrophic strategies. Both mixotrophies yield a synergistic effect in that co-consumption improves efficiency relative to either corresponding sole consumption scenario. This synergy arises from the fact that mixotrophy with either ethylene glycol or methanol enables all assimilated acetate to enter the PHB pathway without carbon loss for energy generation, similarly to acetate-hydrogen mixotrophy (Fig 2B, 2C). On top of that, both ethylene glycol and methanol oxidation yield sufficient excess reducing equivalents to drive some carbon reassimilation - 0.3 mol·mol$^{-1}$ for EG; 0.6 mol·mol$^{-1}$ for methanol (Fig 3B, 3C). As a result of both of these effects, co-consumption with these substrates yields better carbon conservation than in the corresponding sole consumption scenarios.

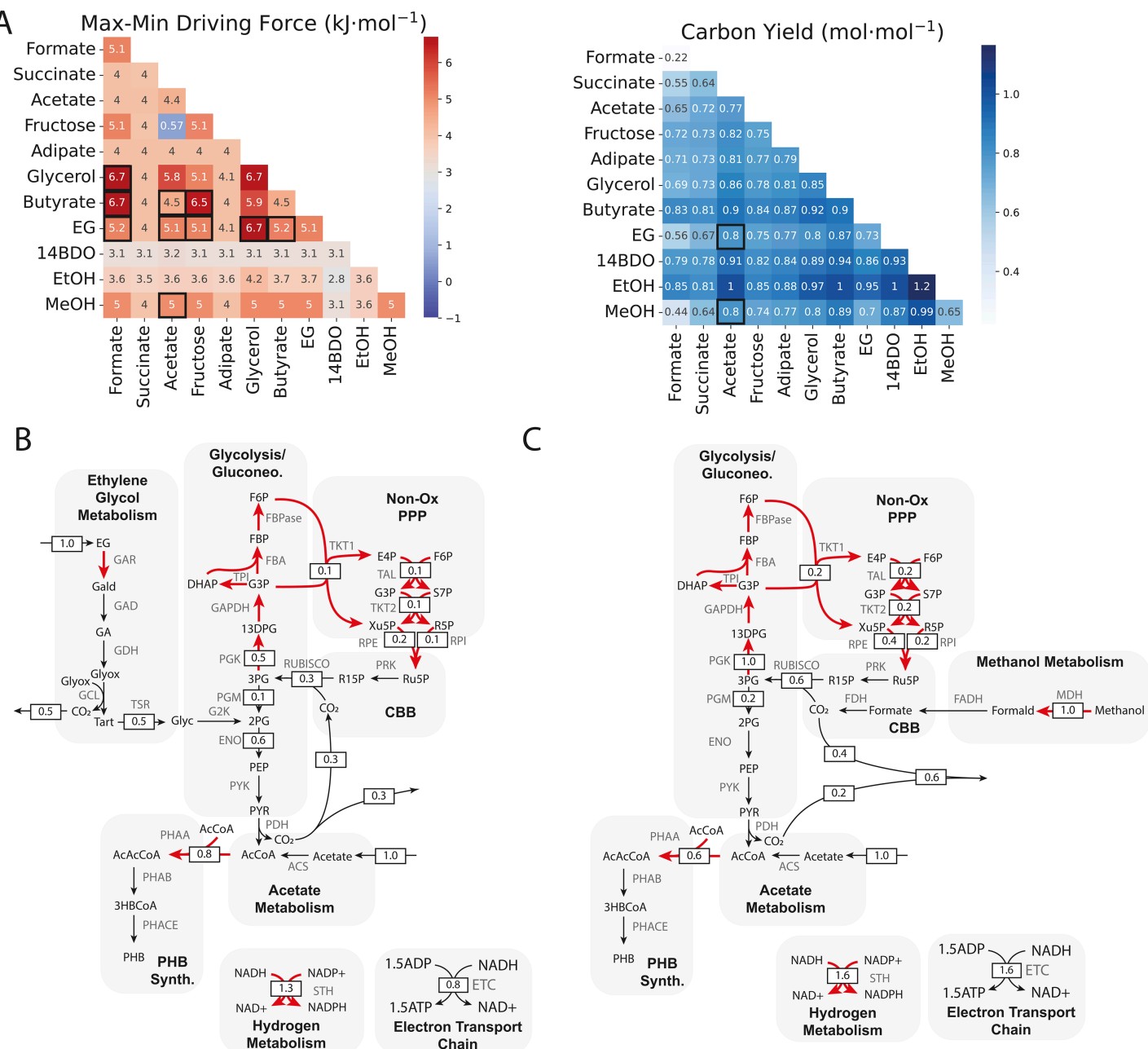

**Fig 3. (A) Mixotrophic performance in terms of MDF (left panel) and carbon yield (right panel) for all carbon source pairings, with sole carbon source scenarios along the diagonals.** Carbon source pairs with MDFs at least as high as the corresponding single carbon source scenarios are boxed in the left panel. Thermodynamically viable pairings with increased carbon yields relative to the corresponding single carbon source scenarios are boxed in the right panel. Two pairings (ethylene glycol [EG]-acetate, methanol [MeOH]-acetate) are both more thermodynamically viable than their corresponding single carbon source scenarios and result in improved carbon yields. (B) Flux distribution for the ethylene glycol-acetate mixotrophy. Ethylene glycol enters central carbon metabolism via 2-phosphoglycerate ('2PG') and/or glyoxylate ('Glyox') following a multi-step oxidation pathway which yields sufficient reducing power to convert acetate to PHB. 0.8 moles of carbon dioxide are lost per mole of ethylene glycol, with 0.5 coming from ethylene glycol oxidation and 0.3 coming from pyruvate dehydrogenase (PDH). Reactions operating at the MDF of 5.1 kJ/mol are highlighted in red. (C) Flux distribution for the methanol-acetate mixotrophy. Methanol is converted to carbon dioxide via three oxidation steps, generating sufficient reducing equivalents for acetate conversion to PHB. 0.6 moles of carbon dioxide are lost per mole of methanol, with 0.4 coming from methanol oxidation and 0.2 coming from PDH. Reactions operating at the MDF of 5.0 kJ/mol are highlighted in red. For both mixotrophies, excess reducing equivalents drive incomplete, but thermodynamically feasible carbon dioxide reassimilation, resulting in improved carbon efficiency compared to that of ethylene glycol, methanol, or acetate as sole carbon sources. GAR: glycoaldehyde reductase; GAD: glycoaldehyde dehydrogenase; GDH: glycolate dehydrogenase; GLC: glyoxylate carboligase; TSR: tartonate semialdehyde reductase; G2K: glycerate-2-kinase; Gald: glycoaldehyde; GA: glycolic acid; Tart: tatronate semialdehyde; Glyc: glycerate; MDH: methanol dehydrogenase; FADH: formaldehyde dehydrogenase; Formald: formaldehyde.

The EG-acetate and methanol-acetate flux distributions, shown in Fig 3B and 3C, also highlight a pattern we observed across mixotrophies. When substrate assimilation generates excess reducing equivalents, the pentose phosphate pathway (PPP) runs as a futile cycle to dump these equivalents via triose phosphate dehydrogenase, which uses an NADPH, and phosphoglycerate kinase, which uses an ATP. As a result, some carbon is lost to handling redox imbalance in addition to that lost as carbon dioxide. Carbon sources that produce sufficient reducing equivalents can reassimilate some carbon dioxide via the complete CBB cycle rather than just the non-oxidative branch of the PPP. Reassimilation is possible for ethanol as a sole carbon source (S3B Fig), and methanol and ethylene glycol in some mixotrophic pairings. In all cases, redox imbalance defines thermodynamic bottlenecks in the metabolic system examined here.

If the co-substrates which are good mixotrophic pairs and can be derived from eCO$_2$R (formate, methanol, ethylene glycol) are considered carbon dioxide-neutral, then the carbon efficiency of PHB production for mixotrophic scenarios changes considerably, as shown in Fig 4. In this case, carbon yields greater than 1 mol·mol$^{-1}$ are possible, because the carbon from the eCO$_2$R-derived co-substrate is "free"; it is essentially a carbon-based electron source replacing hydrogen. Though this is an idealization, it represents the best possible carbon efficiency for the metabolic system under consideration.

In this idealized case, formate can only support 100% carbon yield when it is paired with butyrate, though other viable formate pairings (formate-ethylene glycol, formate-14BDO) have improved carbon yield relative to either sole consumption case. This is because formate can only produce a single reducing equivalent for every carbon dioxide molecule its oxidation yields, so there is a low cap on the carbon yield improvement that is possible with formate as a carbon dioxide-neutral electron source. Butyrate conversion to PHB generates the remaining

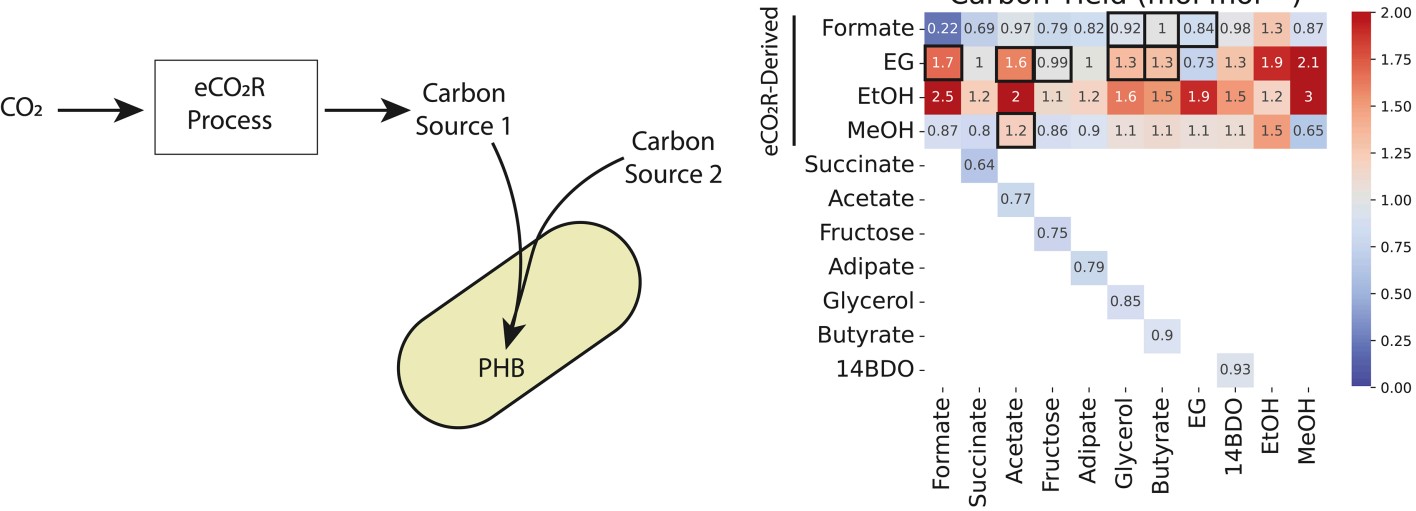

**Fig 4. Carbon yield for mixotrophies with one co-substrate derived from a carbon dioxide-free eCO$_2$R process.** In this hypothetical scenario, carbon dioxide is converted to formate, ethylene glycol (EG), ethanol (EtOH), or methanol (MeOH), which is then co-fed to a mixotrophic strain with another sustainably-derived feedstock. In the ideal case, carbon from the eCO$_2$R-derived co-substrate does not factor into the carbon yield. Yields for thermodynamically feasible pairings (see Fig 3) with one co-substrate being eCO$_2$R-derived are boxed. Carbon dioxide-free ethylene glycol would provide the largest feasible increase to carbon yield, possibly resulting in carbon dioxide-negative conversion of its co-substrates. Yields for sole consumption scenarios are included for comparison.

reducing equivalents required for carbon conservation via butyryl-CoA dehydrogenase when it is paired with formate, enabling complete conversion.

Ethylene glycol supports the most significant carbon yield improvements of all eCO$_2$R-derived electron sources in viable pairings because it contains two carbons and its assimilation yields at least two reducing equivalents. As a result, all thermodynamically viable ethylene glycol mixotrophies have yields of at least 1 mol·mol$^{-1}$ if the ethylene glycol is considered to be carbon dioxide-free. Though eCO$_2$R-derived ethanol mixotrophies have the highest yields, none of these pairings are thermodynamically viable, since ethanol mixotrophies are generally more thermodynamically constrained than consumption of the non-ethanol carbon source alone (Fig 3A).

At present, eCO$_2$R cells tend to produce mixtures of at least a few of the carbon-based electron sources we considered. Thus, it is possible that the carbon yield for a number of these mixotrophic scenarios could be much higher than the values determined here, because the carbon intensity of both co-substrates could be very low. However, this would require the product mixture from the eCO$_2$R to contain carbon sources yielding feasible mixotrophies, which may be challenging given that ethanol mixotrophy is likely to be challenging. In the case in which all carbon is low-carbon dioxide or carbon dioxide-free, a more relevant constraint would be the rate at which conversion is possible in the mixotrophic scenarios. This would require estimating constraints on uptake rates for each mixotrophy, which is beyond the scope of the present work. However, our results do suggest that in hypothetical integrated electrochemical-fermentation processes, it might be worthwhile to "over-reduce" carbon dioxide to target ethylene glycol and/or methanol production, if possible, because co-consumption of these substrates can enable significant improvements in downstream carbon efficiency relative to other, less reduced co-substrates.

## Mixotrophy with partial oxidation is a viable strategy for low-carbon dioxide fermentation

Another strategy to increase carbon yield in carbon-carbon mixotrophies is to partially oxidize the electron-carrying co-substrate into a valuable exported intermediate. This is only possible with C2+ co-substrates with a partially oxidized intermediate that has value as a chemical product. Of the carbon sources investigated here, ethylene glycol and glycerol meet these criteria. Ethylene glycol can be efficiently oxidized to glycolic acid (GA), yielding two reducing equivalents [42]. Glycolate dehydrogenase activity is required for glycolic acid to enter central carbon metabolism, so if this activity is blocked, excess glycolic acid is exported and accumulates externally instead of being assimilated (Fig 5B). Glycolic acid is a valuable commodity chemical that is used to produce a variety of products, such as cosmetics, coatings, adhesives, etc. Therefore, the development of low-carbon dioxide glycolic acid production routes is desirable [43].

A similar motif exists for glycerol, which can be oxidized in a two-step pathway to 3-hydroxypropionate (3HP). 3HP is also a valuable commodity for which sustainable production routes are actively sought [43]. Unlike glycolic acid, 3HP is not known to be assimilated into central carbon metabolism, but glycerol assimilation via glycerol kinase must be blocked for efficient conversion to 3HP instead (Fig 5E). The first step in the glycerol-to-3HP pathway is commonly catalyzed by glycerol dehydratase, which requires cobalamin as a cofactor [44]. Though *C. necator* does not natively produce cobalamin [45], exogenous addition of this vitamin has been used to support heterologous glycerol dehydratase activity [46]. Therefore, we have included the pathway here to show the potential of its application in carbon conserving mixotrophy.

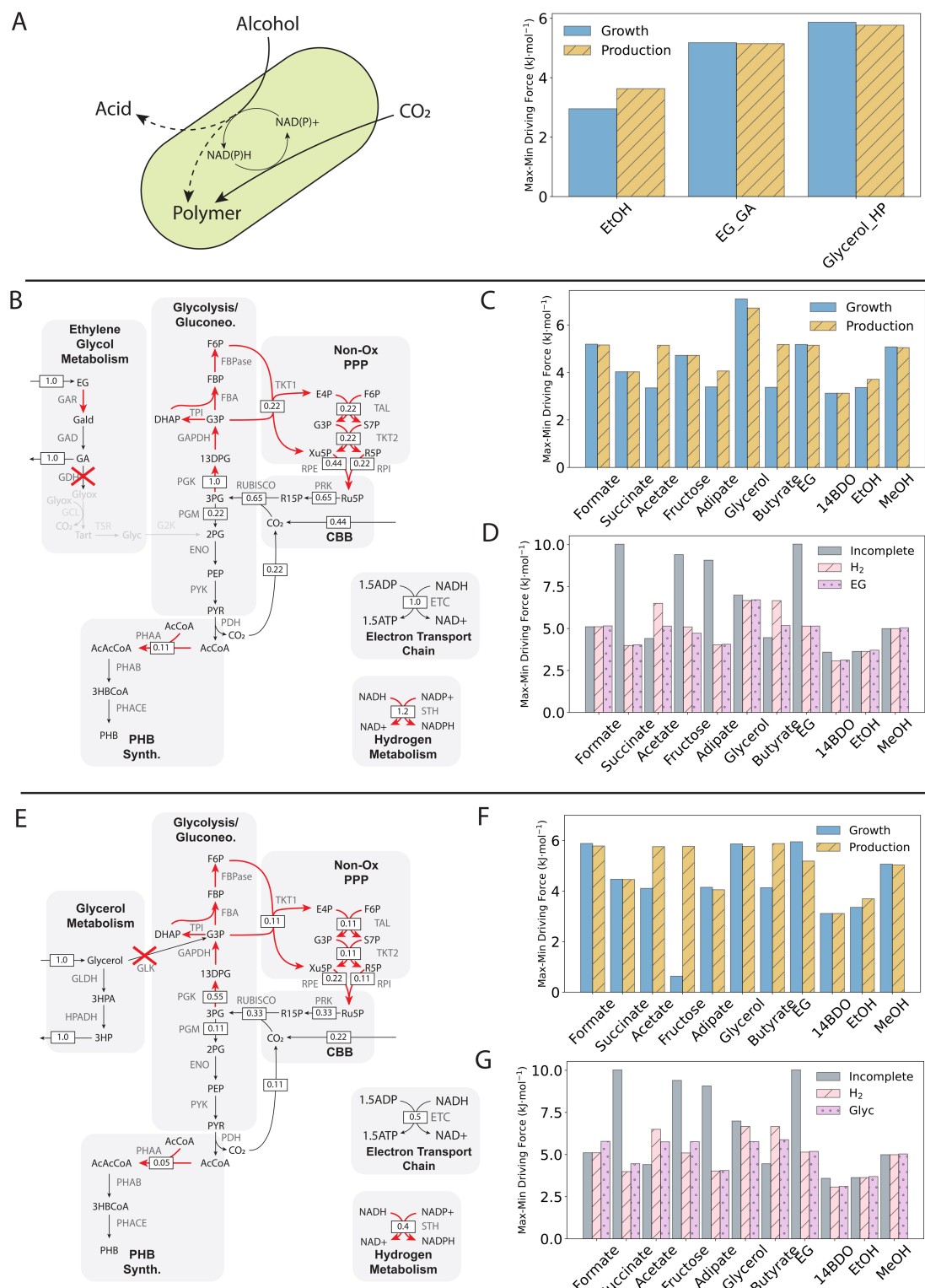

**Fig 5. (A) Alcohols as carbon dioxide-free, carbon-based electron sources.** The partial oxidation of alcohols to exported value-added acids, or biosynthetic intermediates could provide excess reducing equivalents in place of hydrogen, with concomitant carbon dioxide reduction to a biopolymer, such as PHB. Of all carbon sources examined, only ethanol, ethylene glycol

with glycolic acid (GA) export ('EG_GA'), and glycerol with 3-hydroxypropionate (3HP) export ('Glycerol_HP') can feasibly support growth and production without an additional electron source. Oxidation of the alcohol generates sufficient reducing equivalents for carbon dioxide conversion to biomass or PHB. (B) Flux distribution for ethylene glycol partial oxidation with relative fluxes indicated and reactions operating at the MDF of 5.1 kJ/mol highlighted in red. Glycolyate dehydrogenase (GDH) activity must be blocked to prevent ethylene glycol assimilation into central carbon metabolism. The oxidation of a single mole of ethylene glycol to glycolic acid yields sufficient reducing equivalents to drive the transformation of 0.44 moles of carbon dioxide, resulting in 0.11 molecules of PHB, alongside 1 mole of glycolic acid. (C) Thermodynamic feasibility of mixotrophy with ethylene glycol as a source of electrons. For all carbon sources investigated, mixotrophic fermentation is thermodynamically feasible for both growth-only and PHB production-only modes. (D) Thermodynamic feasibility of various carbon utilization scenarios, including incomplete utilization ('Incomplete'), hydrogen-supported complete utilization ('H$_2$'), and complete utilization supported by partial ethylene glycol oxidation ('EG'). Ethylene glycol-supported complete carbon conversion is as thermodynamically feasible as hydrogen-supported complete conversion, except in the case of acetate and butyrate, and is as feasible as incomplete conversion for many of the carbon sources investigated. (E) Flux distribution for glycerol partial oxidation with relative fluxes indicated. Glycerol kinase (GLK) activity must be blocked to prevent glycerol assimilation into central carbon metabolism. The oxidation of a single mole of glycerol to 3HP yields sufficient reducing equivalents to drive the transformation of 0.22 moles of carbon dioxide, resulting in 0.05 moles of PHB produced, alongside 1 mole of 3HP. (F) Thermodynamic feasibility of mixotrophy with glycerol as a source of electrons. For all carbon sources investigated, mixotrophic fermentation is thermodynamically feasible for both growth-only and PHB production-only modes. (G) Thermodynamic feasibility of various carbon utilization scenarios, including complete utilization supported by partial glycerol oxidation ('Glyc'). Similarly to ethylene glycol partial oxidation, glycerol partial oxidation can feasibly support complete carbon conversion for all carbon sources investigated and is as feasible as incomplete conversion for many carbon sources.

We constrained models capable of consuming ethylene glycol and glycerol to enforce partial oxidation and acid product export. We also allowed carbon dioxide uptake in these models, and blocked hydrogen uptake. When these constraints are active, ethylene glycol and glycerol serve solely as electron sources; carbon for growth and production comes from carbon dioxide via the CBB cycle (Fig 5A). We screened all single-substrate models, including these partial oxidation models, for their ability to feasibly support growth and production with hydrogen uptake and carbon dioxide export blocked, but carbon dioxide import allowed. As expected, only the ethanol consuming, and glycolic acid and 3HP producing models can support carbon dioxide-consuming growth and production without electrons from hydrogen. In the glycolic acid model, ∼0.4 moles of carbon dioxide must be assimilated per mole of ethylene glycol oxidized for both growth and production (Fig 5B, S4 Fig), while for the 3HP model, ∼0.2 moles must be assimilated per mole of glycerol oxidized (Fig 5E, S4 Fig). The ethanol, ethylene glycol-to-glycolic acid, and glycerol-to-3HP models all also have thermodynamically viable growth and production modes, as shown by their positive MDFs (Fig 5A).

The ethylene glycol-to-glycolic acid and glycerol-to-3HP pathways could provide the necessary reducing equivalents to drive carbon conservative transformation of waste-derived carbon sources in mixotrophic scenarios. In such scenarios, partial oxidation of ethylene glycol or glycerol would provide the electrons required for co-substrate conversion to PHB, and reassimilation of any carbon dioxide generated from co-substrate oxidation. We therefore examined each mixotrophic pairing with these pathways, with carbon dioxide efflux and hydrogen input blocked. For both carbon sources, we could find flux distributions with positive MDFs for both growth and production modes for all mixotrophic scenarios (Fig 5C, 5F), indicating that such mixotrophies are thermodynamically feasible.

Some mixotrophic scenarios with the glycolic acid model required additional carbon dioxide assimilation beyond reassimilation of carbon generated by oxidation of the co-substrate (S4 Fig). Acetate, butyrate, 14BDO, ethanol, methanol mixotrophies require de novo carbon dioxide assimilation when electrons are provided by the ethylene glycol-to-glycolic acid pathway. This is unsurprising for the acetate and butyrate mixotrophies, since we have already shown that complete ethylene glycol oxidation provides sufficient reducing equivalents for conversion of both to PHB. In the glycolic acid model, no carbon dioxide is generated from

ethylene glycol oxidation, so excess reducing equivalents are not required for carbon reassimilation, and instead drive de novo carbon dioxide assimilation. The glycerol-to-3HP pathway only supports additional carbon dioxide assimilation in the case of ethanol mixotrophy (S4 Fig). Ethanol utilization alone requires carbon dioxide assimilation (S3B Fig), but coupling with both of the acid producing pathways increases the stoichiometric ratio of carbon dioxide uptake relative to the ethanol-only scenario because they provide additional reducing equivalents beyond the excess that ethanol oxidation produces.

To determine the practical viability of glycolic acid and 3HP mixotrophies, we compared their MDFs to those of the single carbon source scenario without complete carbon conversion, and to those of the hydrogen-powered complete carbon conversion scenario (Fig 5D, 5G). All partial oxidation mixotrophies have driving forces which are approximately the same as the case in which hydrogen drives complete carbon conversion ('H$_2$'), because they have the same bottleneck reactions involved in carbon fixation (Fig 5B, 5E). This is not true for acetate and butyrate, in which hydrogen-supported complete conversion is more favourable, because assimilation of these substrates is constrained by ATP recycling (Fig 2C), rather than by NADH:NAD.

These results suggest that mixotrophic fermentation with partial oxidation of either ethylene glycol or glycerol could be a viable approach to carbon dioxide-free upcycling of many of the waste streams examined here. Co-production of an acid renders carbon conservation viable because it provides carbon dioxide-free electrons at the point of fermentation. Though there is a carbon dioxide cost to these electrons over the entire life cycle of the resulting outputs, reduction in carbon intensity at one step in that life cycle may be useful to reduce the overall carbon intensity. Ethylene glycol and glycerol are also more practical electron sources than hydrogen, because they are safer, and more miscible in water. Moreover, the practical feasibility of the acid-polymer co-production scenario is supported by the fact that the acids are exported while the polymer accumulates intracellularly. Biomass separation is a necessary downstream step of all bioprocesses, so recovery of both products is possible within the status quo approaches to downstream processing.

## Discussion

We have shown that there is significant potential to use *C. necator* as a catalyst to harvest electrons from waste-derived feedstocks to drive carbon dioxide utilization. This may provide a novel pathway to reducing the predicted high hydrogen requirement of a future circular carbon economy, while motivating the utilization of recalcitrant waste streams as inputs to chemical production. Critical to this potential is the native ability of *C. necator* to use carbon dioxide in response to excess available reducing equivalents from the oxidation of substrates with high DOR. This allows it to address redox imbalances by reducing carbon dioxide as required to store excess electrons in PHB. To date, this ability has primarily been exploited by feeding *C. necator* long(er)-chain molecules that induce redox imbalance from their consumption as sole carbon sources [31,34]. We have described several configurations which could enable similar utilization of *C. necator*'s native abilities using mixtures of waste streams instead to improve carbon yield, thermodynamic driving forces, and overall sustainability.

Our analysis shows that ethylene glycol is broadly very favourable as a carbon-based electron source because it can be feasibly combined with a number of other waste-derived carbon sources without significantly affecting thermodynamic viability, but while still improving carbon yield. Moreover, its partial oxidation yields a valuable product, but has the same thermodynamic feasibility as status quo hydrogen-coupled carbon fixation. Acetate and butyrate are

also critical to achieving thermodynamically feasible, high carbon yield mixotrophic fermentation, in part because of our choice of product. However, a number of other valuable chemical products, such as long-chain fatty acids, can be derived from acetyl-CoA or butyryl-CoA with energy input via reducing equivalents derived from an additional electron source such as hydrogen. Ethylene glycol mixotrophy and hydrogen-supported conversion of acetate and butyrate should thus be explored further in *C. necator*.

Additionally, the partial oxidation scenarios we examined could yield low-carbon dioxide biopolymers beyond PHB without acid co-production because they generate intermediates which can be co-polymerized with 3-hydroxybutyrate. Both glycolate [47,48] and 3HP [49] co-polymers of 3-hydroxybutyrate have been demonstrated. Thus, while the present work is limited to a single polymer product, our focus on inputs may enable future work which considers production of a wide variety of chemical products using low-carbon dioxide mixotrophies.

We did not consider any metabolic rewiring to improve efficiency in our analysis, because doing so would have expanded the design space to an impractical degree. Yet, improvements to thermodynamic driving forces via rewiring have been shown in *C. necator* [35], and these demonstrated strategies would likely be useful to address the common thermodynamic constraints associated with redox imbalance that we found among most of the metabolic scenarios we investigated. These - and other, as-yet undescribed strategies - could be applied in the mixotrophic scenarios we have evaluated here to further improve carbon conservation and thermodynamic feasibility. The present work may help to limit the design scope for metabolic rewiring approaches, since we have shown non-viability of a number of synthetic mixotrophies. Future work should focus on the viable combinations we have elucidated, namely those making use of acetate and butyrate for carbon and ethylene glycol or methanol for reducing equivalents.

Practically, achieving mixotrophy in a *C. necator* platform - or any microbe, for that matter - will be challenging. In the wild, access to single carbon sources is often limited; a ready supply of two or more at the same time would be exceedingly rare. As such, complex metabolic regulatory systems have evolved to prioritize the use of limited cellular resources for sequential carbon source utilization over co-consumption. These regulatory systems present a significant barrier to achieving the proposed synthetic mixotrophies. Despite these challenges, synthetic, carbon dioxide-utilizing mixotrophy has been demonstrated in *E. coli* previously [50,51]. *C. necator* may be a better platform for the establishment of synthetic mixotrophy than other industrial microbes because it seemingly maintains continual expression of proteins required for assimilation of many different carbon sources in preparation for environmental shifts [36]. However, it is still known to use global regulators to prioritize high-quality carbon sources over low-quality ones [52]. Moreover, regulation of PHB production is known to be influenced by nutrient availability, which may have unintended interactions with the proposed assimilation pathway. Thus, experimental work to understand how global regulators could be manipulated and/or circumvented will likely be required to build strains capable of the mixotrophies proposed here.

## Materials and methods

### Curation of complementary feedstocks

A literature search was performed to establish a list of compounds which could reasonably be used mixotrophically. Complementary feedstocks which could be derived from $eCO_2R$ and/or those which are monomers of plastics and/or those which are byproducts of other bioprocesses were considered. For example, glycerol is a byproduct of biodiesel production

[53], so it was included. Only feedstocks with characterized assimilation pathways could be included. Anaerobic assimilation pathways were excluded. This yielded a list of five alternative feedstocks (ethylene glycol, methanol, adipate, 1,4-butanediol, and ethanol) and six that are commonly been reported for *C. necator* (carbon dioxide, formate, acetate, butyrate, fructose, succinate). All feedstocks are presented in Table 1, along with their sources, assimilation pathways, and references to literature in which these pathways are described.

Degree of reduction (DOR), defined here as the number of electrons per carbon atom which could be released via oxidation was calculated as follows for each carbon source:

$$\text{DOR} = \frac{4n_C + n_H - 2n_O}{n_C} \tag{1}$$

where $n_C$, $n_H$, and $n_O$ are, respectively, the number of carbon, hydrogen, and oxygen atoms in the molecule. The DOR for each carbon source was determined and is similarly presented in Table 1.

## Model building and modification

The core model constructed by Janasch, *et al* [35] was modified to establish a starting point from which mixotrophy-capable models were generated. The original core model does not include hydrogenase activity, so soluble (SH) and membrane hydrogenase (MH) reactions were added. The SH converts hydrogen and NAD+ into NADH and a proton, while the MH converts hydrogen and ubiquinone to ubiquinol. Additionally, the fructose and formate transport reactions in the original model were removed for convenience based on the downstream workflow. Finally, the non-native phosphoketolase and ATP-citrate lyase reactions were kept in the model, but were blocked for all analyses. These reactions were added by Janasch, *et al* to address thermodynamic bottlenecks that they uncovered through their analysis. Since they are not present in wild-type *C. necator*, they were excluded. These modifications yielded a baseline model capable of growing only on carbon dioxide and hydrogen.

From here, uptake and assimilation pathways for all ten carbon sources were individually added to the baseline model using the COBRApy package in Python3 to produce ten unique models capable of assimilating a single carbon source other than carbon dioxide. Multiple models were constructed, rather than a single model capable of using multiple carbon sources, to avoid spurious overlap between assimilation routes. Each model was evaluated for its ability to produce biomass solely using the complementary carbon source to ensure that the curated assimilation pathways were viable. The uptake of the carbon source for each model was constrained to a value of 1 mmol·gCDW$^{-1}$·h$^{-1}$ and hydrogen uptake was blocked, then parsimonious flux balance analysis (pFBA) was used with biomass as the objective function to find the maximum biomass yield, $Y_{X/S}^{max}$. For fructose and acetate sole consumption models, default constraints were used to ensure predicted fluxes reflected biological reality. Phosphoglucoisomerase flux was enforced for fructose to ensure that it was assimilated via the Entner-Doudoroff pathway, and isocitrate lyase flux was enforced in the case of acetate to simulate an active glyoxylate shunt.

## Carbon-hydrogen mixotrophy

Production envelopes were generated by varying the lower bound on the flux through the biomass reaction between 0 and $Y_{X/S}^{max}$ for an uptake rate of 1 mmol·gCDW$^{-1}$·h$^{-1}$ and finding the optimal PHB yield, $Y_{P/S}$, for each intermediate biomass yield. pFBA with the PHB export reaction as the objective function was used to find $Y_{P/S}$ along the production envelope.

This was performed first with hydrogen uptake blocked and carbon dioxide export uncon-strained, then with hydrogen uptake unconstrained, but carbon dioxide export blocked to force complete reassimilation of carbon. The hydrogen requirements for operation at $Y_{X/S}^{max}$ and the maximum PHB yield, $Y_{P/S}^{max}$, were determined by normalizing the predicted SH flux for each carbon source to its carbon number.

The expected hydrogen cost in Fig 1E was determined by:

$$\frac{q_{H_2}}{n_C} = \frac{\text{DOR}_{\text{PHB}} - Y_{\text{NAD(P)H/DOR}} \cdot \text{DOR}_C}{v_{\text{RUBISCO}}} = \frac{4.5 - 0.5\text{DOR}_C}{1.5} \tag{2}$$

Where 4.5 corresponds to the DOR of PHB, $\text{DOR}_{\text{PHB}}$. For each degree of reduction of the carbon source, $\text{DOR}_C$, one fewer half-mole of hydrogen is required to reassimilate carbon dioxide, because the yield of reducing equivalents per DOR, $Y_{\text{NAD(P)H/DOR}}$, is 0.5 (i.e. each DOR can be converted to one half mole of reducing equivalent). The value of the denomi-nator accounts for the fact that carbon dioxide assimilation requires 1.5 mol of flux through RuBisCO, $v_{\text{RUBISCO}}$, for every mole converted to PHB, because pyruvate dehydrogenase pro-duces 0.5 mol of carbon dioxide for every mole of carbon dioxide assimilated.

### Max-min driving force estimation

The maximum-minimum thermodynamic driving force (MDF) was determined using the OptMDFPathway package [40] in Python3 with the pH set to 7.4, pMg at 3, ionic strength at 200 mM, and at a temperature of 30°C. Briefly, finding the MDF for a given set of reactions involving metabolites with log-transformed concentrations, $x_i$, requires solving the linear optimization problem:

$$
\begin{aligned}
\underset{B, \vec{x}}{\text{maximize}} \quad & B \\
\text{subject to} \quad & -\left(\Delta_r \mathbf{G}'^\circ + RT \cdot \mathbf{S}^T \vec{x}\right) \geq B \\
& \ln c_{i,min} \leq x_i \leq \ln c_{i,max} \quad i = 1, \dots, n
\end{aligned}
$$

Where $B$ is the minimum driving force for all reactions in a pathway. The optimized value of $B$ is the MDF, at which multiple reactions in a pathway may operate. $\Delta_r \mathbf{G}'^\circ$ is a vector con-taining the standard Gibbs energy of the pathway reactions, $\mathbf{S}$ is the stoichiometric matrix describing the active reactions of the pathway, $\vec{x}$ is the vector of log-transformed metabolite concentrations, and $c_{i,min}$ and $c_{i,max}$ are the upper and lower bounds on concentrations for metabolite $i$ of $n$ total metabolites involved in the pathway.

The default lower and upper bounds on internal metabolite concentrations were set to $c_{i,min} = 1\ \mu\text{M}$ and $c_{i,max} = 10\ \text{mM}$. Default cofactor constraints in the OptMDFPathway pack-age were modified to allow for NADH:NAD ratios between 0.08 and 0.35 and NADPH:NADP ratios between $1 \cdot 10^{-4}$ and 0.3 based on reported values for *C. necator* operating mixotrophi-cally [34]. Other cofactor constraints were not adjusted. A .csv file containing cofactor bounds is included in the supplemental material. Standard Gibbs free energies for reactions were determined using the component contribution method included in the OptMDFPathway package. Transport reactions were excluded from all MDF calculations for simplicity. Accu-rately capturing transport reactions in the MDF formulation hinges on having good esti-mates of the bounds on concentrations in specific compartments, which is challenging. Thus, these reactions were excluded to avoid unjustified assumptions. Similarly, the biomass and

PHB reactions were excluded because, as terminal reactions that inherently do not operate at steady-state, it is not clear how they can be included in the MDF framework. Biomass and PHB accumulation are known to occur spontaneously, so it was assumed that they are thermodynamically feasible in the conditions examined.

### Carbon-carbon mixotrophic performance

Mixotrophic models were constructed through pairwise addition of consumption modules to each sole carbon source model, without repeat, since the order in which the consumption modules are combined does not matter (e.g. ethylene glycol-acetate is the same as acetate-ethylene glycol). Uptake rates for each carbon source were constrained to a maximum of 1 mmol·gCDW$^{-1}$·h$^{-1}$, such that a total uptake rate of 2 mmol·gCDW$^{-1}$·h$^{-1}$ was possible in each mixotrophic case. All mixotrophic yields were normalized to total carbon uptake to account for this. The yields presented in Fig 3 were determined according to:

$$Y_{\text{PHB/S}} \left[ \text{C-mol/C-mol} \right] = \frac{4 \cdot v_{\text{PHB}}}{n_{C1} \cdot v_{C1} + n_{C2} \cdot v_{C2}} \tag{3}$$

where $v_{\text{PHB}}$ is the flux through the PHB synthesis reaction, $n_C$ is the carbon number of each substrate, and $v_C$ is the uptake flux of each carbon source. When one of the carbon sources was considered carbon dioxide-free, its contribution to the yield was simply omitted.

The same uptake bounds and pFBA optimization approach were used to find MDFs for the mixotrophic models. In the case of the EG-to-GA and glycerol-to-3HP models, the appropriate export and transport reactions of either consumption model were unconstrained, whereas the reactions required for assimilation into central carbon metabolism were constrained to zero flux. pFBA was then used to find optimized flux distributions, as described.

### Code and Data Availability

Consumption modules corresponding to those listed in Table 1 for each carbon source which can yield both biomass and PHB are included in the Supplementary Information, along with the curated core model and cofactor constraint file. All data and code used for analysis and visualization are available at github.com/theL-A-B/mixotrophy.

### Supporting information

**S1 Data. Core metabolic model for _Cupriavidus necator_.**
(XLSX)

**S2 Data. Consumption modules for all non-CO$_2$ carbon sources.**
(XLSX)

**S3 Data. Cofactor concentration constraints for MDF analysis.**
(CSV)

**S1 Fig. (A) Carbon reassimilation via RuBisCO without an additional electron source is detrimental to both biomass and PHB yield.** A fraction of total carbon flux was forced through the RUBISCO reaction with hydrogen uptake blocked and biomass or PHB production as the objective function. Yields decline as the proportion of carbon that passes through RUBISCO increases. (B) Complete carbon conversion with electrons from hydrogen improves yields. For each carbon source, optimal PHB production for a given minimum biomass flux

was determined, both with complete carbon conversion driven by hydrogen oxidation ('Complete', dashed line) and with carbon dioxide efflux allowed ('Incomplete', solid line). Carbon reassimilation expands production envelopes for all carbon sources, yielding stoichiometric conversion to PHB (Fig 1D).
(EPS)

**S2 Fig. Hydrogen savings for PHB production with carbon conservation relative to the case in which all carbon is oxidized to carbon dioxide first, then reduced using hydrogen.** C1 compounds must be assimilated this way, so there are no relative hydrogen savings for these feedstocks. Conversion of acetate, butyrate, and ethanol to PHB requires notably less hydrogen than in the full oxidation case, while ethylene glycol (EG) conversion involves minimal hydrogen savings. 14BDO: 1,4-butanediol; EtOH: ethanol; MeOH: methanol.
(EPS)

**S3 Fig. (A) Flux distributions for butyrate transformation to PHB with incomplete carbon conversion (top panel) and complete carbon conversion (bottom panel) with relative flux indicated.** In the unforced scenario, 11% of assimilated butyrate must be converted to reducing equivalents via the carbon dioxide-producing TCA cycle. The requirement of fumarase ('FUMC') and malate dehydrogenase ('MDH') activities for energy generation yields an MDF of 4.5 kJ/mol. All reactions shown in red operate at this MDF. In the complete conversion scenario, hydrogen oxidation provides sufficient reducing power for all of the assimilated butyrate to be converted to PHB with no carbon dioxide generation, and therefore no reassimilation required. This reduces the thermodynamic constraint, allowing for operation at an MDF of 6.7 kJ/mol. (B) Flux distribution for ethanol (EtOH) utilization. Ethanol oxidation to acetyl-CoA yields sufficient reducing equivalents to drive polymerization and de novo carbon dioxide assimilation via the CBB. Each mole of ethanol used forces assimilation of 0.32 moles of carbon dioxide, yielding 0.58 moles of PHB. Reactions operating at the MDF of 3.63 kJ/mol are indicated in red. EDH: ethanol dehydrogenase; ACEDH: acetaldehyde dehydrogenase; BCL: butyryl-CoA ligase; BCDH: butyryl-CoA dehydrogenase; 3HBDH: (R)-3-hydrobutyryl-CoA dehydratase; BOX: $\beta$-oxidation (lumped); Acetald: acetaldehyde; ButCoA: butyryl-CoA; CrotCoA: crotonyl-CoA.
(EPS)

**S4 Fig. (A) De novo carbon dioxide assimilation is required for ethylene glycol-to-glycolic acid mixotrophies.** Here, ethylene glycol is a source of electrons only, with carbon coming from the co-substrate and/or carbon dioxide. Partial oxidation of ethylene glycol alone ('EG') requires carbon dioxide assimilation. (B) Glycerol-to-3-hydroxypropionate mixotrophies do not require significant de novo carbon dioxide assimiliation. Partial oxidation of glycerol alone ('Glycerol') requires carbon dioxide assimilation.
(EPS)

## Acknowledgments

We acknowledge that this work was carried out on the traditional territory of the Attawandaron, Anishinaabeg and Haudenosaunee peoples. The University of Waterloo is situated on the Haldimand Tract, the land promised to the Six Nations that includes ten kilometers on each side of the Grand River.

## Author contributions

**Conceptualization:** Michael Weldon, Christian Euler.

**Methodology:** Christian Euler.

**Writing – original draft:** Michael Weldon, Christian Euler.

**Writing – review & editing:** Michael Weldon, Christian Euler.

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
