## [Decision Letter · Decision Letter 0]

1 Apr 2025

PCOMPBIOL-D-25-00355

Mixotrophy for carbon-conserving waste upcycling

PLOS Computational Biology

Dear Dr. Euler,

Thank you for submitting your manuscript to PLOS Computational Biology. After careful consideration, we feel that it has merit but does not fully meet PLOS Computational Biology's publication criteria as it currently stands. Therefore, we invite you to submit a revised version of the manuscript that addresses the points raised during the review process.

Please submit your revised manuscript within 60 days Jun 01 2025 11:59PM. If you will need more time than this to complete your revisions, please reply to this message or contact the journal office at ploscompbiol@plos.org. Please include the following items when submitting your revised manuscript:

We look forward to receiving your revised manuscript.

Kind regards,

Zhuangrong Huang, Ph.D.

Academic Editor

PLOS Computational Biology

Mark Alber

Section Editor

PLOS Computational Biology

**Journal Requirements:**

At this stage, the following Authors/Authors require contributions: Christian Euler, and Michael Weldon. Please ensure that the full contributions of each author are acknowledged in the "Add/Edit/Remove Authors" section of our submission form.

5) We have noticed that you have uploaded Supporting Information files, but you have not included a list of legends. Please add a full list of legends for your Supporting Information files after the references list.

6) Please ensure that the funders and grant numbers match between the Financial Disclosure field and the Funding Information tab in your submission form. Note that the funders must be provided in the same order in both places as well.

**Reviewers' comments:**

Reviewer's Responses to Questions

**Comments to the Authors:**

Reviewer #1: Please find my review comments uploaded as an attachment.

Reviewer #2: Summary:

The authors assess the possibility to use additional carbon/energy sources other than CO2 and H2 to produce biomass or PHB in the chemolithoautotroph Cupriavidus necator. The primary motivation is the limitation of H2 as energy source and the possible difficulties to supply it in the required amounts, while alternative energy (and carbon) sources could be generated from waste streams. The authors investigate various scenarios from basic growth on CO2 + H2 to re-assimilation of emitted CO2 using extra reducing power, to various mixotrophic scenarios (carbon source + H2, two carbon/energy sources).

General evaluation:

The manuscript is overall highly interesting and presents results that are very relevant for the metabolic engineering community. The manuscript is well written with some exceptions outlined in the specific comments. The modeling that was used to investigate mixotrophy is appropriate for these questions and the data that is presented is convincing. However, the methods and results presented here are quite complex and the information density is high. The manuscript can certainly benefit from focusing on the main findings and improving the presentation in the figures. Also, ordering and highlighting important findings can be improved as outlined in the comments below.

Comments:

Figure S1A: is it really possible that yield is constant for nearly all substrates regardless how much of it is re-cycled through Rubisco? This feels counter-intuitive, I would expect a similar picture as for fructose, because full oxidation of the C source and re-uptake of CO2 should mean increased energy cost.

Figure 1C: x-axis labels should be shifted to the left such that their ends align better with the bars. This applies to most of the other bar charts as well.

Figure 4: the difference to Figure 3A (yield) is not entirely clear. The only parameter that changes in Figure 4 is that one of the 2 mixotrophic substrates is considered carbon-neutral (or negative), and thus changes the yield for the better? It is questionable if such a small change in the scenario warrants an extra figure, it could be added as a sub-figure to figure 3. Please also use the same color code for the yield heatmaps in figures 3 and 4.

L99: "CO2 flux was costrained to zero" -- there must be CO2 uptake flux in the case of CO2 acting as substrate, consider changing to "CO2 efflux" or "CO2 emission" was constrained

L101: "complete carbon conservation supported by H2 oxidation improves both biomass and PHB yield" -- Figure 1B shows the opposite: dashed line, without reassimilation, shows higher yield for PHB and biomass in all conditions

L162: "PHB production from EG and ethanol is only thermodynamically feasible when RuBisCO flux is allowed" -- this finding is in contrast to the other substrates where Rubisco rather needs to be blocked to obtain thermodynamically feasible driving force. Can the authors speculate why this is different for these two C2 substrates?

L179: "Therefore, for most carbon sources, CO2-free fermentation via carbon reassimilation is unlikely to be practically achievable, even if it may be thermodynamically feasible." -- A highly interesting conclusion; can the authors maybe elaborate what 'thermodynamically unfavorable' would practically mean? Because a yield on substrate benefit through CO2-reassiilation would mean a strong *evolutionary* driving force. But I assume that when the MDF for this pathway is low, the practical consequence would mean increased enzyme cost to catalyze the reaction, and considerable reverse flux?

L204: "Since CO2-free carbon-H2 mixotrophy is generally infeasible for the carbon sources examined here" -- this is a contradiction to the previous paragraph where you actually show that H2-assisted acetate growth is a viable solution. Consider rephrasing.

L269-: In this entire paragraph, the authors jump between figures and results quite a bit. It starts with Figure 5E , then goes to 5B, 5E again and then 5A. Please consider reordering this section such that the reader is guided from figures 5A to 5G in the right order.

For example in line 293: "Remarkably, the MDF for production in both partial oxidation models is

significantly higher than that of the CO2/H2 baseline, even though these models uses the same route for CO2 assimilation." -- what sub-figure results is this part referrring to? The CO2/H2 base line is not included in Figure 5 as far as I can tell.

The entire section about mixotrophy with partial oxidation of EG and Glycerol is difficult to understand, not the least because the scenarios investigated are quite complex (co-feeding EG or glycerol, CO2 or another carbon substrate, with or without H2 supplementation, with or without permission to emit CO2, etc). This section and the corresponding figure 5 could benefit from focusing only on the main findings, improving the links between text and sub-figures, and maybe improving guidance in the figure itself, e.g. having a schematic which carbon and energy sources and emission type is used for which bar chart. The flux maps are generally of high quality but lack this type of guiding information. For example in figure 5B it is clear that EG is a co-substrate and that conversion is blocked at the GDH enzyme. But to which scenarios of the bar chart in Figure 5C do the flux values correspond? Is CO2 fixed (re-assimilated), and does it come from the oxidized carbon substrate? Is H2 used, and where in the flux map would it appear? Colored inputs corresponding to the scenarios in Figure 5C/D could help here.

For the discussion, it might be worthwhile to add a comment regarding toxicity of substrates and products when it comes to the practical implementation of mixotrophic strategies. Likewise, complete flux to product is unrealistic yet used in many of the simulations. The authors should discuss (briefly) if and how the whole cell catalyst could switch from growth to production, and what difficulties could arise from that regarding product yield, viability, etc. This is not the focus of the paper and does not need to be extensive, but it should be made clear that metabolic modeling can be far from reality.

**Have the authors made all data and (if applicable) computational code underlying the findings in their manuscript fully available?**

Reviewer #1: Yes

Reviewer #2: Yes

PLOS authors have the option to publish the peer review history of their article (what does this mean?). If published, this will include your full peer review and any attached files.

Reviewer #1: **Yes: **Markus Janasch

Reviewer #2: **Yes: **Michael Jahn

**Figure resubmission:**
---

## [Decision Letter · Decision Letter 1]

29 Jul 2025

Dear Dr Euler,

We are pleased to inform you that your manuscript 'Mixotrophy for carbon-conserving waste upcycling' has been provisionally accepted for publication in PLOS Computational Biology. I also attach some extra comments from one reviewer which is mainly related to sentences that can be fixed easily. 

Best regards,

Zhuangrong Huang, Ph.D.

Academic Editor

PLOS Computational Biology

Mark Alber

Section Editor

PLOS Computational Biology

Reviewer's Responses to Questions

**Comments to the Authors:**

Reviewer #1: The updated manuscript addresses all major and minor points sufficiently. Nonetheless, I have further comments to parts of the manuscript which I believe could clarify certain results and their description/presentation. This can be done easily with few additional sentences or small fixes.

1. A more detailed explanation as to what the difference between the results displayed in Figure 1C/1E and Figure S2 are.

2. Line 214: Acceptable reasoning regarding efficient protein allocation, but the “cofactor balancing” might need a sentence (or half-sentence) more of an explanation as to how it’s connected to the MDF. Cofactor balancing is usually a stoichiometric issue.

3. Fig 3B and 3C are not referred to in the manuscript text

4. Inconsistencies between line 290 vs 292 vs 304: EG and glycerol are first introduced as carbon sources, then declared “solely as electron sources”, then referred to as carbon sources again.

5. Method section “Carbon-hydrogen mixotrophy” (lines 416-430): If hydrogen cost (Fig 1E and others) is determined via v_Rubisco, but v_Rubisco is no required for acetate, EtOH and butyrate (line 128), how does the resulting number for hydrogen cost comes to be for these compounds?

6. S1B mentioned before S1A

7. I recommend naming of figures in the supplementary information as “S1”, not just “1”, to be consistent with naming in the main manuscript

8. Line 314: Figure 2C does not any longer refer to EtOH, but rather acetate. Please check manuscript thoroughly for further such outdated references from older versions of the manuscript.

Typos and such:

Generally, check manuscript again for typos, missing words, inconsistencies.

- Overall consistency in the text regarding abbreviations vs written (e.g. methanol vs MeOH, or EG vs ethylene glycol)

- Line 191: missing start of sentence (“This”?)

- 240: CBB cycle

- Line 258: “mixotrophies high the highest yields”

- Lines 376 and 377 “it is still known to use”, “is known to influenced by”

Reviewer #2: The authors have addressed the criticism and sufficiently improved the manuscript.

**Have the authors made all data and (if applicable) computational code underlying the findings in their manuscript fully available?**

Reviewer #1: Yes

Reviewer #2: Yes

PLOS authors have the option to publish the peer review history of their article (what does this mean?). If published, this will include your full peer review and any attached files.

Reviewer #1: **Yes: **Markus Janasch

Reviewer #2: **Yes: **Michael Jahn

---

## [Editor Report · Acceptance letter]

PCOMPBIOL-D-25-00355R1

Mixotrophy for carbon-conserving waste upcycling

Dear Dr Euler,

I am pleased to inform you that your manuscript has been formally accepted for publication in PLOS Computational Biology. Your manuscript is now with our production department and you will be notified of the publication date in due course.

With kind regards,

Lilla Horvath
